# STK3/4 Expression Is Regulated in Uterine Endometrial Cells during the Estrous Cycle

**DOI:** 10.3390/cells8121643

**Published:** 2019-12-15

**Authors:** Sohyeon Moon, Ok-Hee Lee, Sujin Lee, Jihyun Lee, Haeun Park, Miseon Park, Eun Mi Chang, Keun-Hong Park, Youngsok Choi

**Affiliations:** 1Department of Stem Cell and Regenerative Biotechnology, Humanized Pig Research Center, Konkuk University, Seoul 05029, Korea; 1004sh.moon@gmail.com (S.M.); lsw46340@naver.com (J.L.); phe325@naver.com (H.P.); 2Department of Biomedical Science, CHA University, Gyeonggi-do 13488, Korea; okhlee@cha.ac.kr (O.-H.L.); sj091895@naver.com (S.L.); pkh0410@cha.ac.kr (K.-H.P.); 3Fertility Center of CHA Gangnam Medical Center, CHA University, Seoul 06135, Korea; C094007@chamc.co.kr (M.P.); emchang@cha.ac.kr (E.M.C.)

**Keywords:** STK3/4, uterine endometrial cells, estrogen, estrous cycle

## Abstract

The uterus is dynamically regulated in response to various signaling triggered by hormones during the estrous cycle. The Hippo signaling pathway is known as an important signaling for regulating cellular processes during development by balancing between cell growth and apoptosis. Serine/threonine protein kinase 3/4 (STK3/4) is a key component of the Hippo signaling network. However, the regulation of STK3/4-Hippo signaling in the uterus is little known. In this study, we investigated the regulation and expression of STK3/4 in the uterine endometrium during the estrous cycle. STK3/4 expression was dynamically regulated in the uterus during the estrous cycle. STK3/4 protein expression was gradually increased from the diestrus stage and reached the highest in the estrus stage. STK3/4 was exclusively localized in the luminal and glandular epithelial cells of the uterus, and phosphorylated STK3/4 was also increased at the estrus stage. Moreover, the increase of STK3/4 expression in uteri was induced by administration of estradiol, but not by progesterone injection in ovariectomized mice. Pretreatment with an estrogen receptor antagonist ICI 182,780 reduced estrogen-induced STK3/4 expression and its phosphorylation. The estrogen-induced STK3/4 expression was related to the increase in phosphorylation of downstream targets including LATS1/2 and YAP. These findings suggest that STK3/4-Hippo signaling acts a novel signaling pathway in the uterine epithelium and STK3/4-Hippo is one of key molecules for connecting between the estrogen downstream signaling pathway and the Hippo signaling pathway leading to regulate dynamic uterine epithelium during the estrous cycle.

## 1. Introduction

The uterus is one of most important reproductive organs for successful pregnancy on mammals [1,2]. Unlike other tissues, the uterus undergoes dynamic cyclic changes in dependence on the estrous cycle. The murine estrous cycle lasts 4 to 5 days with four distinct stages: proestrus, estrus, metestrus, and diestrus [3,4]. The dynamic change of the uterus is regulated by the ovarian steroid hormones such as estrogen and progesterone. Luminal and gland epitheliums of the uterus begin to grow at the stage of proestrus when estrogen increases. The morphological and histological changes of uterine endometrium are required for preparing endometrial receptivity for successful implantation of embryo. Recent studies showed that numerous signaling pathways including WNT and hedgehog signaling were regulated during the estrus cycle. However, the networks were poorly understood in the uterus during the estrous cycle [5].

The Hippo signaling is known as a cellular signaling pathway that controls organ size to regulate cell death and proliferation [6]. The signaling pathway is involved in a series of phosphorylation of targets including STK3/4, a large tumor suppressor kinase 1/2 (LATS1/2), and yes-associated protein (YAP). STK3/4 is known as a Hippo orthologue in mammals [6]. STK3/4 phosphorylates LATS1/2 by external signal. The activated LATS1/2 phosphorylates YAP and TAZ which are the primary transcription effectors of Hippo signaling in mammals. The phosphorylated YAP and TAZ translocate to cytoplasm from nucleus leading to turning off its target gene expression. The expression of the Hippo signaling effector TAZ has been reported in human uterus [7]. However, the expression and regulation of other Hippo signaling factors in the uterus remain unknown. Therefore, we investigated the expression of STK3/4 in the uterus and also examined how its expression is regulated by estrogen and progesterone during the estrous cycle.

## 2. Material and Methods

### 2.1. Animal Management

Mouse experiments were performed on 6- to 8-week-old ICR mice (Orient Bio Company, Seongnam, Korea). Mice were maintained in optimal temperature and light controls and allowed freely to take food and water. Experimental and surgical procedures complied with the Guide for the Care and Use of Laboratory Animals and were approved by the Institutional Agricultural Animal Care and Use Committee of CHA University (Approval No. IACUC180149).

### 2.2. Estrous Cycle Validation and Uterus Sampling

To determine the stages of the mouse estrus cycle, the vaginal smear assay was done as previously described [8]. Vagina was gently washed out with phosphate buffered saline (PBS). The washed vaginal materials were collected and dropped onto Superfrost Plus Stain slides (Thermo Fisher Scientific, Waltham, MA, USA). After drying the slides, vaginal cells were incubated in 50% ethanol, 75% ethanol, and 90% ethanol for 5 min and stained with hematoxylin (Vector Laboratory, Burlingame, CA, USA) and eosin Y (Sigma-Aldrich, St. Louis, MO, USA). After incubating cells in xylene, the slides were covered with Permount Mounting Medium (Thermo Fisher Scientific, Waltham, MA, USA). The stages of the estrus cycle were determined by the cytological features described by Mettus and Rane [4,9,10]. Each uterus in proestrus, estrus, metestrus, and diestrus stages was collected. One of two uterine horns was processed for paraffin block and the other one was used for RNA and protein isolation.

### 2.3. Histological and Immunostaining Analysis

Uteri were fixed in 4% paraformaldehyde for histology and immunostaining. After embedded in paraffin block, uterine blocks were sectioned with 5 μm-thickness using a microtome and put on Superfrost Plus Stain slides (Thermo Fisher Scientific, Waltham, MA, USA). For immunofluorescent assay, the slides were deparaffinized and boiled in sodium citrate buffer (10 mM sodium citrate, 0.05% Tween 20, pH 6.0) for 40 min using antigen retrieval steamer (IHC world, Gyeonggi-do, Korea). The antigen-retrieved slides were washed in PBS-T (PBS containing 0.05% Tween-20) and blocked by incubating in blocking buffer (PBS-T with 4% BSA) for 4 h (h) at room temperature. Then, the slides were treated with the indicated primary antibodies for overnight at 4 °C. After washing with PBS-T, the slides were incubated in blocking buffer containing the secondary antibody conjugated with Alexa-Fluor antibodies 488 or 546 (Thermo Fisher Scientific, Waltham, MA, USA) at room temperature for 1 h. The slides were washed with PBS and treated with DAPI (4′,6-diamidino-2-phenylindole) (Thermo Fisher Scientific, Waltham, MA, USA). After washing, the slides were mounted by treating with Mounting Medium (DAKO, Glostrup, Denmark) and covered with glass coverslip. The expression and localization of proteins were analyzed using confocal microscopy (Leica Microsystems, Wetzlar, Germany). Images were converted to JPEG format at the resolution of 1024 × 1024 pixels. The intensity of positive signals was calculated by the software LAS X (Leica Microsystems, Wetzlar, Germany) and Image J software.

To perform immunohistochemical staining, the deparaffinized slides were treated with 0.3% H_2_O_2_ in distilled water to quench endogenous peroxidase activity before antigen retrieval. After antigen retrieval, the sections were incubated in a blocking buffer (PBS-T with 5% normal serum and 1% BSA) and treated with the indicated antibody followed by HRP-conjugated secondary antibody (Thermo Fisher Scientific, Waltham, MA, USA). After washing, the signals from antibody were acquired by incubating with DAB solution (Vector Laboratories, Burlingame, CA, USA). The slides were counterstained with hematoxylin. The slides were dehydrated and mounted with Permount Mounting Medium (Thermo Fisher Scientific, Waltham, MA, USA).

The primary antibodies used in this study are as follows; anti-STK4 (ab51134, Abcam, Burlington, CA, USA), anti-STK4 phospho-Thr183 (FITC) (orb9297, Biorbyt, Cambridge, UK), anti-STK4 phospho-Thr183/STK3 phospho-Thr180 (#3681, Cell Signaling Technology, Seoul, Korea), anti-LATS1 phospho-Thr1079 (ABIN872292, Antibodies-online, Limerick, PA, USA), anti-YAP (sc-271134, Santa Cruz Biotechnology, Dallas, TX, USA), anti-YAP phospho-Ser127 (orb8478, Biorbyt, Cambridge, UK), and anti-β-Actin (sc-47778, Santa Cruz Biotechnology, Dallas, TX, USA)

### 2.4. Ovariectomy and Hormone Treatments

Mice were ovariectomized to investigate the effects of estrogen and progesterone on the expression of Hippo signaling-related genes in the uterus as previously described [8]. After 14 days of recovery, β-estradiol (E_2_, 200 ng/mouse, Sigma-Aldrich, St. Louis, MO, USA) or progesterone (P_4_, 2 mg/mouse, Sigma-Aldrich, St. Louis, MO, USA) was subcutaneously injected into mice. Mice were sacrificed and the uteri were collected at 0, 1, 3, 16, 12, and 24 h after hormone treatment. Five mice were used in each group. To test estrogen receptor-mediated regulation of gene expression, ovariectomized mice were either injected with E_2_ (200 ng/mouse) or pretreated with estrogen receptor antagonist, ICI 182,780 (ICI, 500 μg/mouse, Sigma-Aldrich, St. Louis, MO, USA) 30 min before E_2_ injection. Sesame oil (100 μL/mouse, Sigma-Aldrich, St. Louis, MO, USA) was used as control.

### 2.5. RNA Preparation, RT-PCR, and qRT-PCR

Total RNAs were isolated from uteri using RNeasy total RNA isolation kit (Qiagen, Hilden, Germany) according to the manufacturer’s instruction. The total RNAs (2 μg) were reverse-transcribed using SuperScript^®^ III First-Strand Synthesis System (Thermo Fisher Scientific, Waltham, MA, USA). The reverse-transcribed cDNAs were used for RT-PCR and quantitative RT-PCR analyses. RT-PCR was performed using C1000 Touch™ Thermal Cycler (Bio-Rad Life Sciences, Hercules, CA, USA). The annealing temperatures used in the thermal cycling were indicated in Table 1. The products were analyzed in 2% agarose gel.

Semi-quantitative RT-PCR analysis was performed using The iQ™ SYBR^®^ Green Supermix (Bio-Rad Life Sciences, Hercules, CA, USA) and the CFX96 Touch™ Real-Time PCR Detection System (Bio-Rad Life Sciences, Hercules, CA, USA). The results were analyzed with the iQ5™ Optical system software (Bio-Rad Life Sciences, Hercules, CA, USA). The relative gene expression was calculated by the comparative CT (ΔΔCT) method [11]. The relative gene expression was compared with the amounts of either ribosomal protein L-7 (*Rpl7*) or *Gapdh* as reference genes.

### 2.6. Knockdown of STK4 Expression in Human Uterine Endometrial Cells

To examine the effect of *STK4* knockdown on gene expression in human endometrial cells, Ishikawa cell line was used. Ishikawa cells were transfected with *STK4* siRNAs (SR415716, Dharmacon, Lafayette, CO, USA) or universal scrambled negative control siRNA (SR30004, Dharmacon, Lafayette, CO, USA) using Lipofectamine 2000 (Thermo Fisher Scientific, Waltham, MA, USA).

### 2.7. Western Blot and Statistics

Total proteins from cells or uteri were separated on 10% SDS-PAGE and transferred to polyvinylidene difluoride (PVDF) membrane. After blocking with ProNATM phospho-block solution (TransLab, Seoul, Korea), the membrane was incubated with the indicated primary antibody diluted in ProNATM phospho-block solution (TransLab, Seoul, Korea) at 4 °C for overnight. The membrane was treated with HRP-conjugated secondary antibody (OriGene Technologies, Rockville, MD, USA) in ProNATM phospho-block solution to detect protein expression. The immunoreactive bands were detected by chemiluminescence using ECL Western Blotting substrate kit (GenDEPOT, Barker, TX, USA). The relative expression was imaged by ChemiDoc XRS system (Bio-Rad Life Sciences, Hercules, CA, USA) and analyzed by One-way ANOVA analysis.

## 3. Results

### 3.1. Expression of STK3 and STK4 in the Mouse Uteri during the Estrous Cycle

To investigate the expression of *STK3* and *STK4* in the uterus, we evaluated the relative level of *STK3* and *STK4* transcripts during the estrous cycle using RT-PCR and qRT-PCR. As shown in Figure 1A, *STK3* and *STK4* expression was dynamically regulated during the estrous cycle divided into four stages; proestrus, estrus, metestrus, and diestrus. First, both *STK3* and *STK4* transcripts were highly increased at the estrus stage and the increase was significantly reduced at the stage of metestrus (Figure 1B). And then the reduction was rebound at the diestrus stage. The differential STK3/4 expression during the estrous cycle was confirmed by western blotting analysis (Figure 1C). The level of STK3/4 protein expression showed similar pattern with that of their mRNAs. The expression of STK3/4 protein remained relatively high in diestrus, proestrus, and estrus phase of the estrous cycle, whereas it was decreased in the metestrus. This implies that the regulation of *STK3* and *STK4* expression is related to the estrous cycle. Vaginal smear assays confirmed each stage of the estrous cycle (Figure 1D).

In the next study, we looked into the location and expression level of STK3/4 and phosphorylated STK3/4 (pSTK3/4) in the uterus during the estrous cycle. STK3/4 is a cytoplasmic kinase which acts as the first factor in the core Hippo signaling pathway [6]. If the Hippo signal turns on, STK3/4 is activated by phosphorylation [12]. Immunofluorescent images revealed that the location of STK3/4 was dynamically regulated during the estrus cycle. STK3/3 was exclusively located in luminal epithelium (LE) and glandular epithelium (GE) of uterus at diestrus, proestrus, and estrus stages of the estrous cycle (Figure 1E). The relative signal of STK3/4 was highest at the estrus and was reduced at the metestrus (Figure 1E). Next, we investigated the status of STK3/4 phosphorylation during the estrus cycle. As shown in Figure 1E, the level of pSTK3/4 signal was also high at the estrus and was reduced at the metestrus (Figure 1E). This indicates that Hippo signaling through STK3/4 activation as well as its expression regulation occurs rapidly in the uterus during the estrous cycle.

### 3.2. Hormonal Regulation of STK3/4 Expression in Mouse Uterus

We examined whether STK4 expression is affected by two major female steroid hormones, estrogen or progesterone, which are involved in uterine dynamics during the estrous cycle. We used ovariectomized mice for checking the response of STK4 expression by administration of estrogen or progesterone. RT-PCR analysis showed that the expression of *STK4* transcripts was increased in time-dependent manner after administration of estrogen (Figure 2A). The *Stk4* expression was highest at 6 h after administration of estrogen (Figure 2B). We checked the expression of dexamethasone-induced RAS-related protein 1 (*Rasd1*) and lactoferin (*Lf*) to confirm estrogen response in ovariectomized mice. *Rasd1* and *Lf* were known as either early- or late-response genes, respectively [2]. Progesterone treatment didn’t affect the expression of *Stk4* in ovariectomized mice (Figure 2C–D). Progesterone effectiveness in uterus was confirmed by progesterone-induced expression of progesterone-responsive genes such as amphireguline (*Areg*), and *Hoxa10* (Figure 2C,D).

Next, the expression of STK3/4 and its phosphorylation was observed hourly after hormone administration in OVX mice. Immunofluorescent staining with anti-STK3/4 and anti-pSTK3/4 antibodies revealed that STK3/4 and pSTK3/4 expressions were increased by estrogen in LE and GE of uterus (Figure 2E). The level of STK4 was highest at 6 h after treatment of estrogen and then decreased. This suggests that STK4 expression and activation are controlled by estrogen during the estrous cycle.

### 3.3. Estrogen-Dependent Expression of STK3/4 is Mediated via Estrogen Receptor

In order to determine whether estrogen-induced *Stk4* expression is mediated by estrogen receptor, the ovariectomized mice were pretreated with an estrogen receptor antagonist ICI 182,780 (ICI) 30 min before estrogen injection. The expression of *STK4* transcript and STK3/4 protein was examined 6 h after estrogen administration. As shown in Figure 3A,B, estrogen-induced expression of *STK3/4* transcript and protein was dramatically blocked by ICI pretreatment. Immunostaining showed that estrogen increased the STK3/4 expression in LE and GE and induced phosphorylation of STK3/4 (Figure 3C), but the increase in levels of STK3/4 and pSTK3/4 was markedly prevented by ICI (Figure 3C). This result suggests that estrogen induces the expression of STK3/4 as well as phosphorylated forms of STK3/4 by estrogen receptor-mediated pathway in LE and GE of the uterus.

### 3.4. Regulation of Downstream Factors in the Hippo Signaling Pathway in the Uterus

The pSTK3/4 immediately phosphorylates LATS1/2 resulting in the Hippo signaling activation [13]. The activated LATS1/2 (pLATS1/2) phosphorylates a downstream target, YAP and the phosphorylated YAP (pYAP(S127)) is sequestered in the cytoplasm leading to preventing transcriptions of targets [14]. Therefore, we investigated downstream signaling of the Hippo pathway in the uterus of ovariectomized mice. First of all, we examined the effect of estrogen on pSTK3/4 and the continuous downstream targets, LATS1/2 and YAP. Western blotting analysis revealed that pSTK3/4, pLATS1/2, and pYAP were immediately increased after E_2_ administration (Figure 4A). Moreover, immuntostaining with anti-pSTK3/4, anti-LATS1/2, anti-YAP antibodies showed that pLATS1/2 and pYAP(S127) were exclusively co-located with pSTK3/4 in LE and GE of the ovariectomized mouse uterus after E_2_ treatment (Figure 4B,C). These increases were prevented by pretreatment of estrogen receptor antagonist, ICI (Figure 4B,C). This suggests that the Hippo signaling pathway is associated with estrogen receptor-mediated signaling in the uterus.

In the previous study, we proved that estrogen signal in uterine epithelium was mediated by p38 mitogen-activated protein (MAP) kinase pathway [2]. Therefore, we examined the relationship between Hippo signaling and non-genomic estrogen signaling pathway. We co-treated a pyridinyl imidazole inhibitor, SB203580 (SB, Sigma-Aldrich, USA), which is widely used to block p38 MAP kinase pathway [15]. As shown in Figure 4D, pSTK3/4 and pYAP(S127) were significantly reduced by inhibition of p38 pathway. This suggests that estrogen-associated p38 intracellular signaling is closely to the Hippo signaling in the uterus.

### 3.5. Knockdown of STK3/4 in a Human Endometrial Cell Line, Ishikawa Cell

In the following study, we investigated the effect of STK3/4 knockdown on the expression of Hippo signaling target genes in a human endometrial cell line, Ishikawa cells. Knockdown efficiency of STK3/4 using siRNA in Ishikawa cells was shown in Figure 5A. Western blot analysis confirmed the cocktail of siRNA for STK4 knockdown almost completely blocked STK3/4 expression. STK3/4 knockdown affected the phosphorylation of downstream of Hippo signaling factors (Figure 5A). The phosphorylated forms of STK3/4, LAST1/2, and YAP were significantly reduced in STK3/4 knockdown cells (Figure 5A).

Next, we examined the expression of various known targets regulated by the hippo signaling pathway including *VEGFA* [2], Krüppel-like factor (*KLF5*) [16], baculoviral IAP repeat-containing 5 (*BIRC5*) [17], connective tissue growth factor (*CTGF*) [18], *E2F* [19], and catenin beta-1 (*CTNNB1*) [19]. RT-PCR and qRT-PCR analysis showed that STK4 knockdown significantly increased the expressions of *VEGFA* and *CTGF*, whereas it reduced *KLF5* expression (Figure 5C). However, the knockdown did not affect on expression of *E2F*, *CTNNB1*, and *BIRC5* (Figure 5C). These results of this study imply that the expression of target genes is temporally and spatially regulated depending on the tissue.

## 4. Discussion

In this report, we investigated the regulation of Hippo signaling pathway in the uterus. We found that the expression of STK3/4 is dynamically changed in the uterine endometrium during the estrous cycle (Figure 1). In addition, we demonstrated that the regulation of its expression proceeds through estrogen and its receptor (Figure 2 and Figure 3). In addition, we showed that the cascade reaction of Hippo pathway was triggered in uterine epithelium to regulate its downstream targets (Figure 4 and Figure 5), suggesting the Hippo signaling pathway plays a role in uterine dynamics during the estrous cycle.

The Hippo signaling pathway is originally found in *Drosophila* and is evolutionarily conserved in mammals. The Hippo signaling pathway is one of well characterized cellular signaling pathways in various cells and tissues [6]. The pathway plays a role in limiting organ size by restraining cell proliferation, inducing apoptosis, and regulating fates of stem/progenitor cells [20,21]. It contains a series of kinase cascade including STK3/4-LATS1/2-YAP/TAZ factors. Recently, several studies have been reported the role of Hippo signaling pathway in uterine diseases. Song et al. suggested that YAP promotes cell proliferation of endometrial stroma in endometriosis [22]. Zhan et al. reported that a component of the Hippo signaling pathway, TAZ was predominantly located in the nucleus of the endometrioid adenocarcinoma tissue [23]. YAP and TAZ are the downstream effector of the pathway and interact with transcription factors such as TEAD family members. These suggest that abnormal expression and location of YAP or TAZ result in human endometrial diseases. Chen et al. reported that Hippo signaling downstream effector, YAP, induces the decidualization of stromal cells in uterine endometrium [24], implying that the normal regulation of Hippo signaling pathway plays a role of uterine dynamics for successful pregnancy. However, there was little known how the Hippo signaling pathway is regulated in the normal uterus until now. The Hippo pathway contains two core kinases including STK3/4 and LATS1/2 [25]. STK3/4 phosphorylates and activates the protein kinase LATS1/2, which is subsequently able to phosphorylate the transcriptional coactivator YAP or TAZ [6]. The phosphorylated YAP (S127) becomes inactive and translocates from nucleus to cytoplasm resulting in turning off or reducing target gene expression. These cascade reactions might be very important for regulation of cellular processes in tissues such as the uterus.

Interestingly, we found that LATS1/2 phosphorylation in the uterine endometrium seems to be faster than phosphorylation of STK3/4 signaling (Figure 4). It might be a kind of technical problem. But there is also a possibility to be existing unknown signaling pathway which is related to LATS1/2 phosphorylation uterine dynamics during the estrous cycle. Recent two reports identified that MAP4Ks (mitogen-activated protein kinase kinase kinase kinase) including MAP4K1/2/3/4 and MAP4K4/6/7 are involved in direct phosphorylation of LATS1/2 [21,26]. Meng et al. proved that MAP4K4/6/7 (homologs of *Drosophila* Happyhour) knockout reduced YAP phosphorylation more significantly than STK3/4 knockout in human cell line [21]. This is consistent with our results showing YAP phosphorylation which is parallel to LATS1/2 phosphoryation (Figure 4). This implies that STK3/4 and MAP4Ks play together in LATS1/2 regulation in the uterine endometrium (Figure 6).

The uterine endometrium has cyclic changes during the estrous cycle for successful pregnancy. It is known that the dynamic change of the uterus is regulated by ovarian steroid hormones, estrogen and progesterone. However, it was not enough to explain the complicated regulation of uterine endometrium during processes. In addition to hormone regulation, other signals need to be associated. In the previous studies, we showed that estrogen could be associated with rapid expression of other signaling molecules such as dexamethasone-induced RAS-related protein 1 (RASD1) and aminoacyl-tRNA-synthetase-interacting multifunctional protein-1 (AIMP1) [2,8]. RASD1 is known as a member of RAS family which signals various cellular processes [27]. Its expression and regulation was mediated by membrane bound estrogen receptor-dependent intracellular signaling pathway rather than classical nuclear receptor [2]. Especially, the induction of *Rasd1* was mediated via p38-mitogen-activated protein kinase (p38-MAPK) signaling in endometrium of mouse uterus [2]. And we found that RASD1 was reduced in the uterine endometrium of patient with repeated implantation failure [28]. Estrogen levels were lower in the RIF patients group. AIMP1 is also known as a multiple player in various cellular processes including immune reaction, angiogenesis, and fibroblast proliferation. We demonstrated that estrogen directly regulates its expression in the uterine endometrium [8]. This finding indicates that estrogen and various signaling molecules are closely related for endometrial cellular changes during the estrous cycle and hormones. In this study, we firstly found that the regulation of STK4/Hippo by estrogen and the estrous cycle. It gives us a new signaling factor for uterine signaling network.

The uterus circulates repeatedly through the three processes of proliferation, differentiation, and degeneration by steroid hormones until menopause. The uterine cells in the endometrium are the most active undergoing the cellular proliferation, differentiation (decidualization), and degeneration in response to the hormones [29]. The uterine dynamics is regulated by complex networks of numerous signals as well as two ovarian hormones, estrogen and progesterone. The Hippo signaling control starting with STK4 regulation seems to be a great candidate in the dynamics during the estrous cycle. STK4 was highly and rapidly expressed by estrogen receptor dependent pathway in both the luminal epithelium and the glandular epithelium (Figure 2). We demonstrated that the regulation of STK4 and its downstream targets was mediated by estrogen receptor dependent intracellular signaling, p38-MAPK. This suggests that the Hippo signaling pathway is connected to estrogen signaling via p38-MAPK pathway (Figure 6).

In conclusion, the present study demonstrated that a key Hippo signaling factor, STK4 is regulated by estrogen in the uterus and presents a novel potential signaling pathway for regulating uterine dynamics during the estrous cycle. The present report provides experimental evidence to support the further investigation of Hippo signaling factors including STK4 as a mediator in estrogen signaling. It will give us better understanding for complicated signaling networks in the uterine environment during the estrous cycle.

## Figures and Tables

**Figure 1 cells-08-01643-f001:**
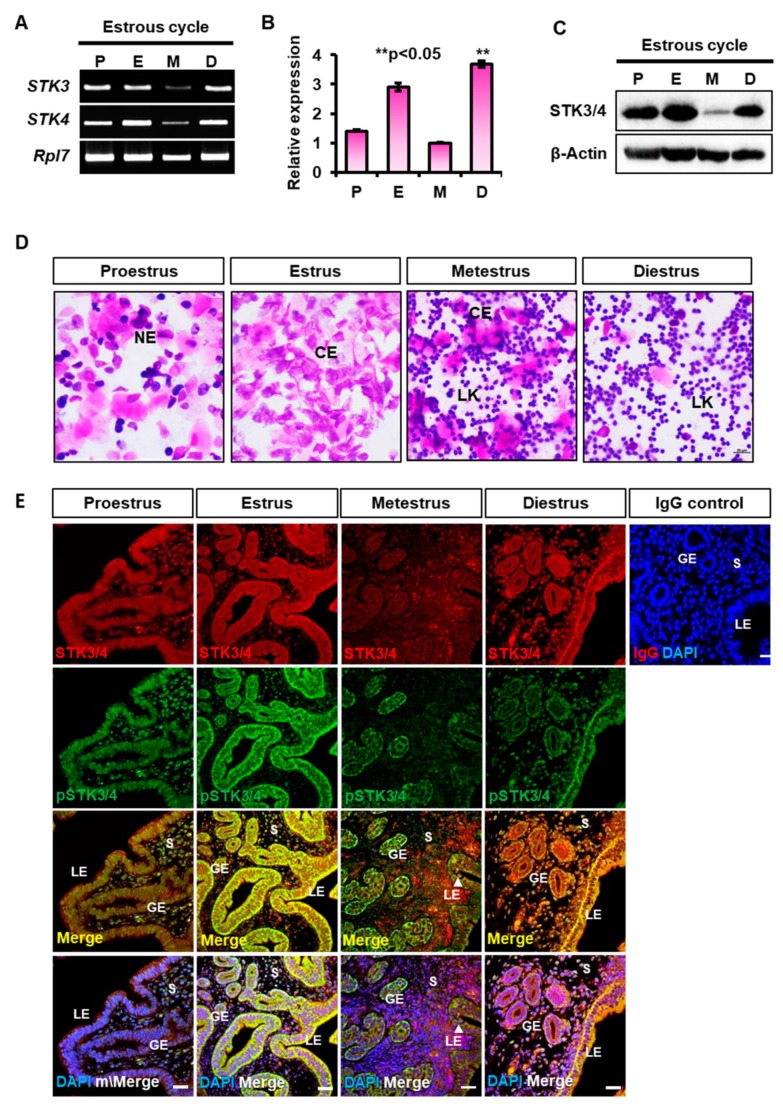
Expression of *STK3* and *STK4* in the mouse uteri during the estrous cycle. (**A**,**B**) The total RNA was isolated from the tissues of 7-week-old mice. RT-PCR and qRT-PCR analysis for *STK3* and *STK4* transcripts in the mouse uteri at four stages of the estrous cycle (P, proestrus; E, estrus; M, metestrus; D, diestrus). Relative expression level of *Stk4* was normalized with *Rpl7* transcript. Data were shown with mean ± SEM. *p*-value; ** *p* < 0.05. (**C**) Western blot analysis of STK3/4 protein was performed using whole cell lysate from mouse uteri during estrous cycle. (**D**) Vaginal smear assays confirming each stage of the estrous cycle. LK, leukocyte; NE, nucleated epithelial cells; CE, cornified epithelial cells. (**E**) Immunohistochemical analysis of STK3/4 and phosphorylated STK3/4 (pSTK3/4) in the 7-week-old mouse uteri at different stages during the estrous cycle. Negative control image is a proestrous uteri stained using normal rabbit IgG (IgG control). LE, luminal epithelium; GE, glandular epithelium; S, stroma. Images were analyzed using a confocal microscope. White bar represents scale bar (scale bar; 25 μm).

**Figure 2 cells-08-01643-f002:**
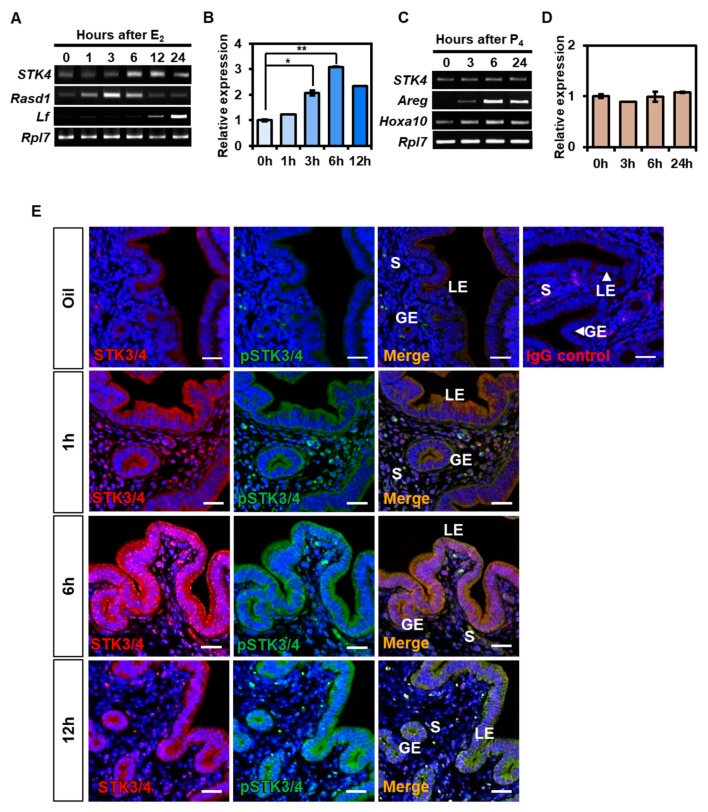
Regulation of STK3/4 expression by estrogen and progesterone in ovariectomized mouse. (**A**,**B**) RT-PCR and qRT-PCR analysis for relative levels of *STK4* transcript in uteri of ovariectomized mice after estrogen (E_2_, 200 ng) treatment. Data were collected at 0, 1, 3, 6, 12, and 24 h after E_2_ treatment. Hormonal responsiveness to E_2_ were confirmed by evaluating levels Dexamethasone-induced RAS-related protein 1 (*Rasd1*) and lactoferrin (*Lf*). (**C**,**D**) Relative expression level of *STK4* transcript in uteri of ovariectomized mice after administration of progesterone (P_4_, 2 mg). Hormone responsiveness to P_4_ were referenced by evaluating levels of amphiregulin (*Areg*) and *Hoxa10*. Relative expression level of *STK4* was normalized with *Rpl7* transcript. Data were shown with mean ± SEM. p-value; * *p* < 0.01, ** *p* < 0.05, (E) Confocal microscopic images represent the localization and expression level of STK3/4 (red) and pSTK3/4 (green) in the uteri of ovariectomized mice treated with E_2_ (200 ng). Uteri were collected at 0, 1, 6, and 12 h after E_2_ administration. Normal rabbit IgG (IgG control) was used as control antibody on uterus sections from mice treated with E_2_ for 6h. Scale bar; 25 μm. Nuclear DNA was stained with DAPI (blue). Images were analyzed using a confocal microscope.

**Figure 3 cells-08-01643-f003:**
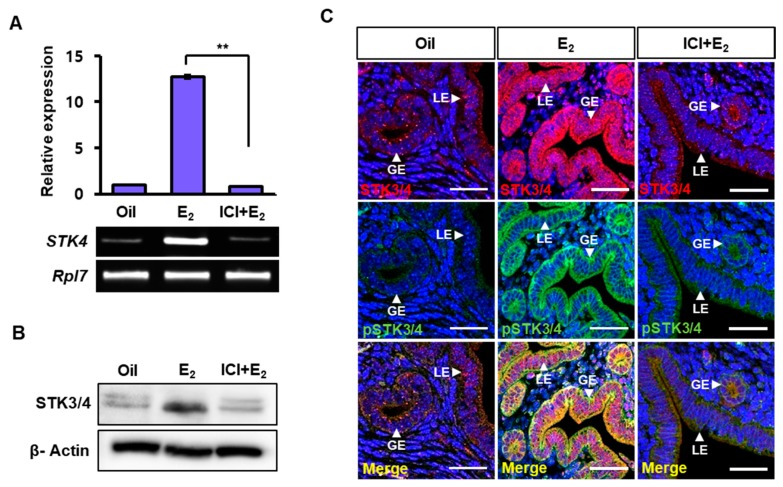
Regulation of *STK3/4* expression through estrogen receptor. (**A**,**B**) Ovariectomized mice were treated with an estrogen receptor antagonist, ICI 182,780 (ICI, 500 μg), for 30 minitues (min) before estrogen (E_2,_ 200 ng) injection and sacrificed at 6 h after E_2_ administration. *STK4* expression was analyzed by RT-PCR and western blot analysis. Data were shown with mean ± SEM. *p*-value; ** *p* < 0.01 (**C**) Immunostaining images showed localization of STK3/4 (red) and pSTK3/4 (green) in uteri of ovariectomized mice treated with oil, E_2,_ or a combination of E_2_ and ICI. LE, luminal epithelium; GE, glandular epithelium. Nuclear DNA was stained with DAPI. The scale bars indicate 25 μm.

**Figure 4 cells-08-01643-f004:**
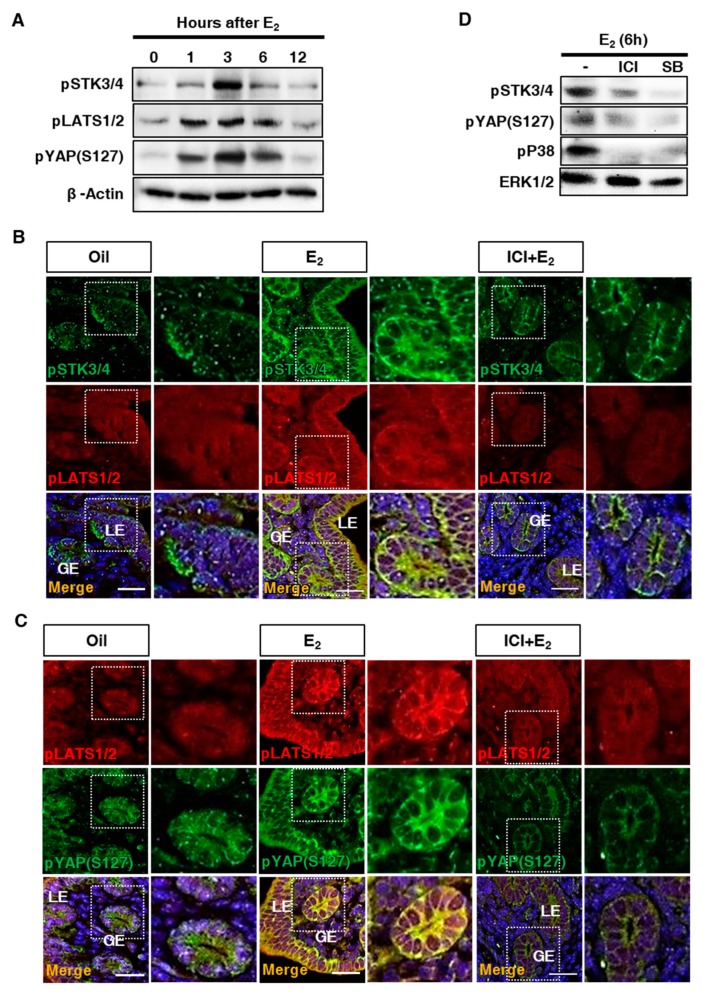
Regulation of LATS1/2 and YAP by estrogen and Hippo signaling pathway in the uterus. (**A**) Western blot analysis for relative levels of STK3/4 downstream signaling factors, STK3/4 phosphorylation (pSTK3/4), LATS1/2 phosphorylation (pLATS1/2) and YAP phosphorylation on Serine 127 (pYAP (S127)) in uteri of ovariectomized mice after estrogen (E_2_, 200 ng) treatment. Data were collected at 0, 1, 3, 6, and 12 h after E_2_ treatment. Beta-actin (β-Actin) was used for an internal control. (**B**) Co-immunostaining images showed localization of pSTK3/4 (green) and pLATS1 (red) in uteri of ovariectomized mice treated with oil, E_2,_ or a combination of E_2_ and an estrogen receptor antagonist, ICI 182,780 (ICI, 500 μg). (**C**) Co-immunostaining images showed localization of pYAP (S127) (green) and pLATS1/2 (red) in uteri of ovariectomized mice treated with oil, E_2,_ or a combination of E_2_ and ICI. LE, luminal epithelium; GE, glandular epithelium. Nuclear DNA was stained with DAPI. The scale bars indicate 25 μm. (**D**) Western blot analysis showed pSTK3/4, pYAP (S127), pP38, and ERK1/2 in uteri of ovariectomized mice treated with E_2_ or E2 in combination with ICI or p38 MAP kinase inhibitor, SB203580 (SB).

**Figure 5 cells-08-01643-f005:**
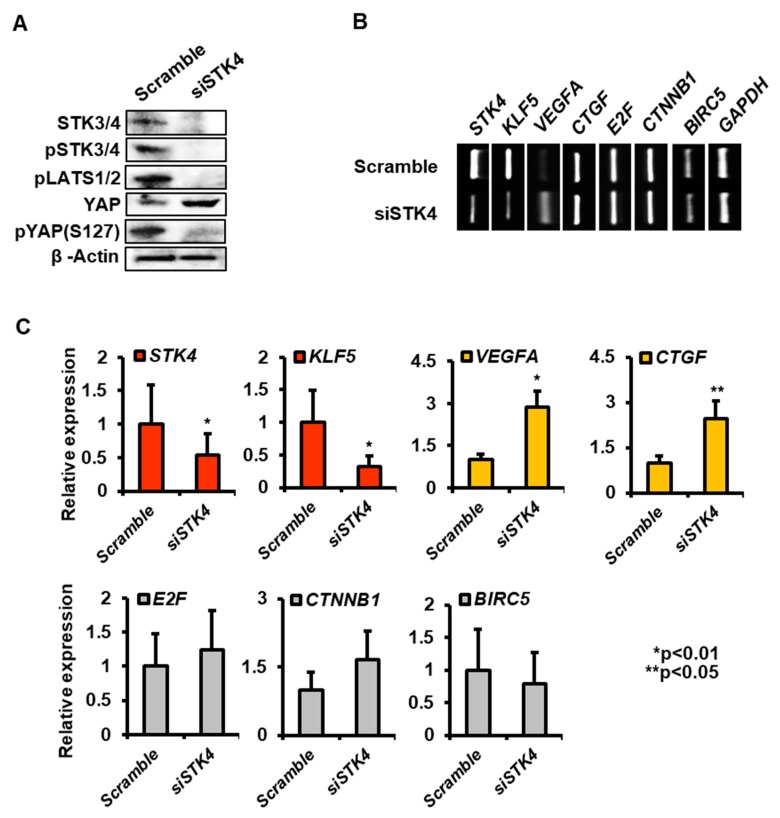
Gene expression by STK3/4 knockdown in Ishikawa cells. (**A**) Western blot analysis showed the expression of STK3/4, pSTK3/4, pLATS1, YAP, and pYAP(S127) in Ishikawa cells treated with either 10 μM of scramble siRNA (Scramble) or STK4 siRNA (siSTK4). (**B**,**C**) RT-PCR and qRT-PCR analysis showed the effect of STK3/4 knockdown on the expression of target genes (*KLF5, VEGFA, CTGF, E2F, CTNNB1,* and *BIRC5*) regulated by Hippo signaling pathway. *GAPDH* level was used as an internal control.

**Figure 6 cells-08-01643-f006:**
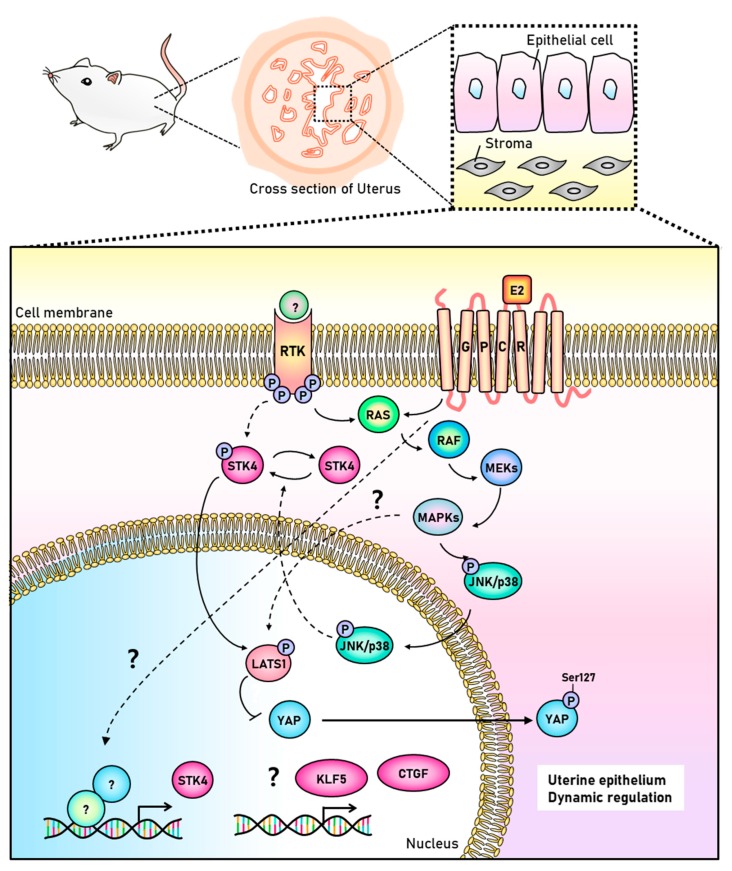
Illustration of cross-talk between the Hippo signaling and estrogen receptor mediated intracellular signaling. The Hippo signaling pathway includes a series of phosphorylation in STK3/4-LATS1/2-YAP axis. The signal of activation of STK3/4 is unknown in endometrial epithelium. The activation of the non-genomic pathway of membrane-bound estrogen receptor might be involved in the activation of STK3/4. The MAP4K pathway is one of candidates to activate LATS1/2 via unknown signaling.

**Table 1 cells-08-01643-t001:** Primer sequences and RT-PCR conditions.

Genes	Accession No.	Primer Sequence	Annealing Temperature (°C)	Product Size (bp)
*Stk4*	NM_021420.3	^(1)^ Fwd; 5′-CAGCCCGAGGAAGTGTTTGA-3′ ^(2)^ Rev; 5′-CACGGGCACTTGCTTGATTG-3′	60	117
*Stk3*	NM_019635.2	^(1)^ Fwd; 5′-TGGTGAAGAGTCCTGAGCAG-3′ ^(2)^ Rev; 5′-CCACGCTTTCTGAACTCGTC-3′	60	228
*Rpl7*	NM_011291.5	^(1)^ Fwd; 5′-TCAATGGAGTAAGCCCAAAG-3′ ^(2)^ Rev; 5′-GAAGAGACCGAGCAATCAAG-3′	60	246
*Lf*	NM_008522.3	^(1)^ Fwd; 5′-AGGAAAGCCCCCCTACAAAC-3′ ^(2)^ Rev; 5′-GGAACACAGCTCTTTGAGAAGAAC-3′	58	141
*Rasd1*	NM_009026	^(1)^ Fwd; 5′-GATGTGCCCAAGCGACTCT-3′ ^(2)^ Rev; 5′-TGAGGAAGCGCGACACAAT-3′	60	110
*Areg*	NM_009704.4	^(1)^ Fwd; 5′-GGTCTTAGGCTCAGGCCATTA-3′ ^(2)^ Rev; 5′-CGCTTATGGTGGAAACCTCTC-3′	60	137
*Hoxa10*	NM_008263.3	^(1)^ Fwd; 5′-CCTGCCGCGAACTCCTTTT-3′ ^(2)^ Rev; 5′-GGCGCTTCATTACGCTTGC-3′	60	203
*Gapdh*	BC092294	^(1)^ Fwd; 5′-AGGTCGGTGTGAACGGATTTG-3′ ^(2)^ Rev; 5′-TGTAGACCATGTAGTTGAGGTCA-3′	60	123
*STK4*	NM_006282.3	^(1)^ Fwd; 5′-CACCCATTTGTCAGGAGTGC-3′ ^(2)^ Rev; 5′-GAACCATCGTGCCAGAATCC-3′	65	163
*VEFGA*	NM_001171623.1	^(1)^ Fwd; 5′-GGCCAGCACATAGGAGAGAT-3′ ^(2)^ Rev; 5′-ACGCTCCAGGACTTATACCG-3′	65	155
*KLF5*	NM_001730.4	^(1)^ Fwd; 5′-ACCCTGCCAGTTAACTCACA-3′ ^(2)^ Rev; 5′-ACCAGGGTAATCGCAGTAGT-3′	65	102
*BIRC5*	NM_001012271.1	^(1)^ Fwd; 5′-AGGACCACCGCATCTCTACAT-3′ ^(2)^ Rev; 5′-AAGTCTGGCTCGTTCTCAGTG-3′	60	118
*CTGF*	NM_001901.2	^(1)^ Fwd; 5′-TCTTCGGTGGTACGGTGTAC-3′ ^(2)^ Rev; 5′-TGTCTTCCAGTCGGTAAGCC-3′	63	249
*E2F*	NM_005225.2	^(1)^ Fwd; 5′-CTTCGTAGCATTGCAGACCC-3′ ^(2)^ Rev; 5′-AAAACATCGATCGGGCCTTG-3′	65	137
*CTNNB1*	NM_001904.3	^(1)^ Fwd; 5′-CTTACACCCACCATCCCACT-3′ ^(2)^ Rev; 5′-CCTCCACAAATTGCTGCTGT-3′	65	197
*GAPDH*	AY340484.1	^(1)^ Fwd; 5′-ATGGGGAAGGTGAAGGTCG-3′ ^(2)^ Rev; 5′-GGGTCATTGATGGCAACAAT-3′	60	107

^(1)^ Fwd; forward primer, ^(2)^ Rev; reverse primer.

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
