# Peer review of "STK3/4 Expression Is Regulated in Uterine Endometrial Cells during the Estrous Cycle"

_cells, 2019, doi:10.3390/cells8121643_

Round 1
Reviewer 1 Report
The authors first demonstrated that Stk4 expression in uterus changes depending on the estrous cycle and that total and phosphorylated Stk4 increase at the estrus. They next showed that estrogen, but not progesterone, enhances Stk4 expression in ovariectomized mice via estrogen receptor. Estrogen also increases phosphorylated Lats1 and Yap. The effect of estrogen is blocked by the inhibition of p38 pathway. Stk4 silencing reduces phosphorylated Lats1 and Yap and accordingly increases Yap target genes such as Ctgf.
This is a descriptive paper but is probably the first report to discuss Stk4 in uterus. It is potentially interesting. But exactly for this reason, I would like to raise the following comments.
Major comments
In this study, the authors focus on Stk4 and Lats1 and do not mention Stk3 or Lats2. The sequences around phosphorylation sites are highly conserved. I am wondering whether or not the antibodies they used cross-react Stk3 and Lats2 proteins. They should clearly explain this point, if the antibodies are specific for Stk4 and Lats1 proteins. Anyway, they should show Stk3 at least in Fig. 1A and then focus on Stk4. They argue that Stk4 expression and activation are enhanced at the estrus in Fig. 1 and by estrogen in Fig. 2. But to conclude that the activation is enhanced, they need to show that the ratio of phosphorylated Stk4 over total Stk4 is enhanced. I rather suspect that the increase of phosphorylated Stk4 simply reflects the increase of total Stk4. In Section 3.3, they show that estrogen receptor antagonist ICI 182,780 blocks estrogen-mediated enhancement of Stk4. Therefore, the effect of estrogen is mediated by estrogen receptor. In the legend to Fig. 6, they describe “The activation of the non-genomic pathway of membrane-bound estrogen receptor might be involved in the activation of STK4”. Does ICI 182,780 inhibit the binding of estrogen to non-genomic receptor too? If this is the case, it would better to add explanation. Or do they mean that estrogen enhances Stk4 expression through both genomic and non-genomic pathways? It might be straightforward if they use specific antagonists against genomic and non-genomic estrogen receptors. 6 shows how Stk4 is activated but does not show how Stk4 mRNA is enhanced. What is the mechanism? Readers want to know this point. Even if it is out of their scope, the authors should discuss it. The result shown in Fig. 4A is inconsistent with the authors’ argument. Lats1 and Yap1 are phosphorylated before Stk4. How do they explain this discrepancy? What is the physiological meaning of the activation of the Hippo pathway at the estrus? They analyzed Yap target genes only in the estrogen-induced model. Are Yap target gene products indeed suppressed at the estrus in mice? What will happen if they knock out Stk4 in uterus? If they want to focus on cell experiments, they could show that estrogen suppresses cell proliferation and induces apoptosis in Ishikawa cells.
Minor points.
The numbers of figures in the text are wrong in many parts. 4D does not show pAkt, although it is mentioned in the text and the legend.. There are grammatical errors in the text.Author Response
We appreciate the reviewer's comments.
Please an attached file including detail responses for them.
Best regards,
Youngsok Choi

Reviewer 2 Report
The study of Moon et al. examined the expression of the Hippo signalling pathway, in special the expression of STK4, in the estrous cycle. The results are novel, the experimental design is straight forward.
Nevertheless the following points have to be adressed:
The figure legend does not always fit with the presented figures. The authors have to pay much more attention to the correctness of their figure legends. Figure 3 presents subpanel A,B and C. The figure legend describes Figure 3A-E. pSTK4 which is mentioned in the figure legend is not shown as western blot in the figure itself.
Figure 4: The cellular localization of pYAP, pSTK4 and pLATS1 is difficult to see in the IF photographs. An additional magnification (maybe as inlet) could clarify the localization of these molecules. pSTK4 shown in Figure 4A is not mentoned in the figure legend. Figure 4D presents a western blot for pSTK4, pYAP, pP38 and ERK1/2. The figure legend describes several more proteins. The authors have to include the missing western blots in Figure 4D.
Figure 5: I would suggest to show the gelphotographs in the same order in Figure 5B like they are presented in Figure 5C. The bars of the relative expression of VEGFA and CTGF in Figure 5C looks rather equal. However the bands of the gelphotographs looks quite different and does not represent the same correlation of VEGFA and CTGF. The authors have to explain this difference and have to clarify this in their manuscript.
How does pSTK4 correlates with STK4 in the estrous cycle? A western blot for pSTK4 has to been shown in Figure 1C.
Does STK4 regulation or the Hippo signalling pathway has any known effects in the dysregulation of the estrous cycle or on tumor development. What could be the benefit of these results for manipulation/therapy of the estrous cycle or endometrial cancer?
Author Response
We appreciate the reviewer's comments.
Please an attached file including detail responses for them.
Best regards,
Youngsok Choi

Round 2
Reviewer 1 Report
I am still concerned about the specificities of anti-MST1 and anti-LATS1 antibodies. First of all, although the authors say “The information was described on materials and methods in the manuscript”, I cannot find any explanation about the specificities. In terms of anti-LATS1 phospho-Thr1079 antibody, the company says “This sequence will cross react with LATS2 if phosphorylated at Thr1041”. Likewise, Abcam says “although they expect that ab51134 does not cross-react with MST2, they have never experimentally checked the specificity.” They should show the evidence that the antibodies (ab51134, orb9292, and ABIN872292) specifically detect STK4 and LATS1. I do not understand why they intentionally avoid showing the result regarding STK3. If STK3 behaves differently from STK4, the finding is interesting and intriguing. Exactly that is why they should show the result. Moreover, they used anti-pSTK4 antibody to evaluate the activity of STK4 in Figure 4 and Figure 5. I believe that this antibody is the antibody #3681 from Cell Signaling Technology, which also detects phospho-STK3. They need to clearly demonstrate whether they indeed study STK4 alone or STK3/4, especially if they consider that STK3 and STK4 are significantly different from each other in uterus. I requested the authors to evaluate the ratio of phospho-STK4 signal over the total STK4 signal. Otherwise, they cannot conclude whether STK4 is activated or whether STK4 expression is simply increased in response to E2. 4A clearly indicates that LATS is activated before STK4 in the uterus. There may be a certain mechanism, by which LATS is activated independently of STK4. If they can examine whether LATS is activated in response to E2 in STK4-depleted Ishikawa cells, it would be ideal. But as they are interested in STK4 in uterus, this is out of their scope, I agree. However, they should discuss the apparent inconsistency in Discussion. Actually, they currently devote much space to the review of the Hippo pathway and the signals in uterus. They should discuss their own results in details.Author Response
I am still concerned about the specificities of anti-MST1 and anti-LATS1 antibodies. First of all, although the authors say “The information was described on materials and methods in the manuscript”, I cannot find any explanation about the specificities. In terms of anti-LATS1 phospho-Thr1079 antibody, the company says “This sequence will cross react with LATS2 if phosphorylated at Thr1041”. Likewise, Abcam says “although they expect that ab51134 does not cross-react with MST2, they have never experimentally checked the specificity.” They should show the evidence that the antibodies (ab51134, orb9292, and ABIN872292) specifically detect STK4 and LATS1. I do not understand why they intentionally avoid showing the result regarding STK3. If STK3 behaves differently from STK4, the finding is interesting and intriguing. Exactly that is why they should show the result. Moreover, they used anti-pSTK4 antibody to evaluate the activity of STK4 in Figure 4 and Figure 5. I believe that this antibody is the antibody #3681 from Cell Signaling Technology, which also detects phospho-STK3. They need to clearly demonstrate whether they indeed study STK4 alone or STK3/4, especially if they consider that STK3 and STK4 are significantly different from each other in uterus.
Response: We agreed with the reviewer’s comment on Stk4 and Lats1. In fact, we didn’t know the meaning between Stk3 and Stk4. But, we performed RT-PCR for STK3 and STK4. We put it on Figure 1A. Their expression patterns look similar. This suggests that the reviewer’s point was right. Therefore, we rewrote the manuscript in terms of Stk3/4 and Lats1/2 instead of Stk4 and Lats 1. We are sorry for the confusion on the words.
I requested the authors to evaluate the ratio of phospho-STK4 signal over the total STK4 signal. therwise, they cannot conclude whether STK4 is activated or whether STK4 expression is simply increased in response to E2. 4A clearly indicates that LATS is activated before STK4 in the uterus. There may be a certain mechanism, by which LATS is activated independently of STK4. If they can examine whether LATS is activated in response to E2 in STK4-depleted Ishikawa cells, it would be ideal. But as they are interested in STK4 in uterus, this is out of their scope, I agree. However, they should discuss the apparent inconsistency in Discussion. Actually, they currently devote much space to the review of the Hippo pathway and the signals in uterus. They should discuss their own results in details.
Response: We tried to western blot analysis to quantitate the ratio of pSTK3/4 over total STK3/4. But the antibody didn’t work on western. That’s why we showed only immunostaining. In terms of LATS1/2 expression, we agreed with the reviewer’s comment. LATS1/2 phosphorylation seems to earlier than STK3/4 phosphorylation. In fact, we were also curious about the time lab. As reviewer’s pointout, there might be independent pathway on LATS1/2 activation. We were describing the inconsistency in Discussion as follows; Interestingly, we found that LATS1/2 phosphorylation in the uterine endometrium seems to be faster than phosphorylation of STK3/4 signaling (Fig. 4). It might be a kind of technical problem. But there is also a possibility to be existing unknown signaling pathway which is related to LATS1/2 phosphorylation uterine dynamics during the estrous cycle. Recent two reports identified that MAP4Ks (mitogen-activated protein kinase kinase kinase kinase) including MAP4K1/2/3/4 and MAP4K4/6/7 are involved in direct phosphorylation of LATS1/2 [21,26]. Meng et al. proved that MAP4K4/6/7 (homologs of Drosophila Happyhour) knockout reduced YAP phosphorylation more significantly than STK3/4 knockout in human cell line [21]. This is consistent with our results showing YAP phosphorylation which is parallel to LATS1/2 phosphoryation (Fig. 4). This implies that STK3/4 and MAP4Ks play together in LATS1/2 regulation in the uterine endometrium (Fig. 6).
Reviewer 2 Report
The authors adressed most of my concerns. There is still one question which was not adressed:
How does pSTK4 correlates with STK4 Expression in the estrous cycle? A western blot for pSTK4 has to be shown in Figure 1C.
Author Response
How does pSTK4 correlates with STK4 Expression in the estrous cycle? A western blot for pSTK4 has to be shown in Figure 1C.
Response:
We tried to western blot analysis to quantitate the ratio of pSTK3/4 over total STK3/4. But the antibody against pSTK3/4 didn’t work on western analysis.That’s why we showed only immunostaining. If you have any good suggestion, please let me have anyone. We need to have good antibody. And also, other reviewer pointed out that we have to use STK3/4 instead of STK4 alone. So we changed it into STK3/4. In fact, we are working on further studies using knockout animal model. We are able to show more solid data on the hippo signaling in the uterine system.